# Urinary Biomarkers in Interstitial Cystitis/Bladder Pain Syndrome and Its Impact on Therapeutic Outcome

**DOI:** 10.3390/diagnostics12010075

**Published:** 2021-12-29

**Authors:** Hung-Yu Lin, Jian-He Lu, Shu-Mien Chuang, Kuang-Shun Chueh, Tai-Jui Juan, Yi-Chang Liu, Yung-Shun Juan

**Affiliations:** 1School of Medicine, College of Medicine, I-SHOU University, Kaohsiung 84001, Taiwan; ed100464@edah.org.tw; 2Division of Urology, Department of Surgery, E-Da Cancer & E-Da Hospital, Kaohsiung 82445, Taiwan; 3Emerging Compounds Research Center, Department of Environmental Science and Engineering, College of Engineering, National Pingtung University of Science and Technology, Pingtung 91201, Taiwan; toddherpuma@yahoo.com.tw; 4Department of Urology, College of Medicine, Kaohsiung Medical University, Kaohsiung 80708, Taiwan; u9181002@gmail.com (S.-M.C.); spacejason69@yahoo.com.tw (K.-S.C.); 5Graduate Institute of Clinical Medicine, College of Medicine, Kaohsiung Medical University, Kaohsiung 80708, Taiwan; 6Department of Urology, Kaohsiung Municipal Ta-Tung Hospital, Kaohsiung 80145, Taiwan; 7Department of Medicine, National Defense Medical College, Taipei 114201, Taiwan; terry870921@gmail.com; 8Division of Hematology-Oncology, Department of Internal Medicine, Kaohsiung Medical University Hospital, Kaohsiung 80708, Taiwan; 9Faculty of Medicine, College of Medicine, Kaohsiung Medical University, Kaohsiung 80708, Taiwan; 10Regenerative Medicine and Cell Therapy Research Center, Kaohsiung Medical University, Kaohsiung 80708, Taiwan

**Keywords:** bladder, interstitial cystitis, bladder pain syndrome, biomarker

## Abstract

Interstitial cystitis/bladder pain syndrome (IC/BPS) is defined as a chronic bladder disorder with suprapubic pain (pelvic pain) and pressure and/or discomfort related to bladder filling accompanied by lower urinary tract symptoms, such as urinary frequency and urgency without urinary tract infection (UTI) lasting for at least 6 weeks. IC/BPS presents significant bladder pain and frequency urgency symptoms with unknown etiology, and it is without a widely accepted standard in diagnosis. Patients’ pathological features through cystoscopy and histologic features of bladder biopsy determine the presence or absence of Hunner lesions. IC/PBS is categorized into Hunner (ulcerative) type IC/BPS (HIC/BPS) or non-Hunner (nonulcerative) type IC/BPS (NHIC/BPS). The pathophysiology of IC/BPS is composed of multiple possible factors, such as chronic inflammation, autoimmune disorders, neurogenic hyperactivity, urothelial defects, abnormal angiogenesis, oxidative stress, and exogenous urine substances, which play a crucial role in the pathophysiology of IC/BPS. Abnormal expressions of several urine and serum specimens, including growth factor, methylhistamine, glycoprotein, chemokine and cytokines, might be useful as biomarkers for IC/BPS diagnosis. Further studies to identify the key molecules in IC/BPS will help to improve the efficacy of treatment and identify biomarkers of the disease. In this review, we discuss the potential medical therapy and assessment of therapeutic outcome with urinary biomarkers for IC/BPS.

## 1. Definition, Diagnostic Criteria and Prevalence of IC/BPS

Interstitial cystitis/bladder pain syndrome (IC/BPS) is defined as a chronic bladder disorder characterized with suprapubic pain (pelvic pain; bladder pain) and pressure and/or discomfort related to bladder filling, which are accompanied by lower urinary tract symptoms, such as urinary frequency and urgency without evidence of urinary tract infection (UTI) last for at least 6 weeks [1,2]. Clinical manifestations of IC/PBS can overlap with those of symptoms such as overactive bladder (OAB), recurrent UTI, chronic pelvic pain syndrome, chronic urethral syndrome, vulvodynia, prostatitis in men, and endometriosis in women [3]. Symptoms of IC/BPS patients involve chronic pelvic pain, often coexisting with insomnia, depression, anxiety, and sexual dysfunction, therefore resulting in impaired quality of life [4] and withdrawal from social activities [5]. The subjective perception on patients as pelvic pain is the distinguishing characteristic for IC/PBS [6]. According to patients’ pathological features through cystoscopy and histologic features of bladder biopsy to determine the presence of Hunner lesions, IC/PBS can be categorized into Hunner (ulcerative) type IC/BPS (HIC/BPS) or non-Hunner (non-ulcerative) type IC/BPS (NHIC/BPS) [7,8]. Nevertheless, the pathophysiology of IC/BPS remained unclear, so the phenotypic classification of IC/BPS has not been defined yet.

The prevalence of IC/BPS increased with age [5,9,10]. It ranged from 2.70% to 6.53% in the American population [1]. In Asian countries, the prevalence was 0.045% in female patients and the male-to-female ratio was 1:5.8 in Japan [11]. In Korea, the prevalence of IC was 0.026% in female patients [1]. In Taiwan, the Taiwan National Database in 2013 revealed that the prevalence of IC/BPS was 0.022%. Among them, the incidence was 0.016% for ages under 40 years, 0.063% between 40 and 65 years, and the incidence increased to 0.086% for age above 65 years, respectively, where the male to female ratio was 1:10 [10].

## 2. Sex Difference in Females and Males with IC/BPS

Urological chronic pelvic pain syndrome (UCPPS) referred to chronic pain in the pelvis, prostate, bladder, and/or genitalia. UCPPS included IC/BPS in females attributed to the bladder as well as chronic prostatitis/chronic pelvic pain syndrome (CP/CPPS) in males. CP/CPPS is defined as chronic genitourinary pain in the absence of uropathogenic bacteria localized to the prostate gland [12]. Clemens et al. indicated variation in the incidence and severity of bladder symptoms using the multidisciplinary approach to the study of chronic pelvic pain (MAPP) database in comparison with females and males with UCPPS [13]. Moreover, females with IC/BPS had significantly worse frequency, urgency and nocturia based on the Interstitial Cystitis Symptom Index (ICSI), Interstitial Cystitis Problem Index (ICPI), and American Urological Association Symptom Index (AUASI), as compared to males with CP/CPPS [13]. Marszalek et al. also found a greater prevalence of storage urinary symptoms in females compared to males according to the International Prostate Symptom Score (IPSS) [14]. IC/BPS has been considered a syndrome mostly affecting females [10].

Previous findings indicated that pain severity was similar in both sexes. Females with UCPPS had greater prevalence of urinary disorders/symptoms than males with UCPPS; symptoms such as frequency, nocturia, and urgency were more often shown in females [15]. Furthermore, most female participants indicated that pubic pain was the most bothersome symptom [15]. Different symptom patterns and clinical phenotypes suggested that there were probably different etiologies and pathogenic pathways between different sexes [15].

## 3. Classification and Pathophysiology of IC/BPS

### 3.1. Classification

The Study of Interstitial Cystitis (ESSIC) subtype patients with BPS into grade 1 (normal), grade 2 (with glomerulations grade II (large submucosal bleeding) or grade III (diffuse global mucosal bleeding)), and grade 3 (Hunner lesions (with or without glomerulations)) according to cystoscopy with hydrodistension, and classified into grade A (normal), grade B (with inconclusive), and grade C (histology showing inflammatory infiltrates and/or detrusor mastocytosis and/or granulation tissue and/or intrafascicular fibrosis) according to biopsy diagnosis [16]. The European Association of Urology (EAU) guidelines further give a recommendation that grade A diagnosis requires hydrodistension and biopsy [17]. Clinically, IC/BPS could be classified into IC/BPS with Hunner lesions (HIC/BPS) or without Hunner lesions (NHIC/BPS) through cystoscopy and histologic features of bladder biopsy [18]. The prevalence of Hunner ulcer was found about 6–8%, which was associated with severe symptom and profound reduced functional and anesthetic bladder capacity [19,20]. Clinical characteristic differences between HIC/BPS and NHIC/BPS are shown in Table 1. However, the etiology and pathogenesis of IC/BPS remained obscure.

### 3.2. The Etiology and Pathogenesis of IC/BPS

Not only urothelium, but also detrusor muscle, peripheral afferent terminals, and pelvic blood vessels all played an important role on underlying pathophysiological mechanism of IC/PBS. Urothelial cells expressed numerous receptors/ion channels, including receptors for adenosine, norepinephrine, acetylcholine, neurotrophins, endothelins, and various transient receptor potential (TRP) channels [21]. Release of chemical mediators from urothelial cells could regulate intercommunication with afferent and efferent nerves, adjacent urothelial cells, or other cells (e.g., myofibroblasts and immune or inflammatory cells) within the bladder wall. The bladder lamina propria is composed of an extracellular matrix containing a variety of cells, such as mesenchymal cells, fibroblasts, interstitial cells, and sensory nerve endings (afferent nerve) and also contains a rich vascular and lymphatic system, elastic fibers, and muscularis mucosae. However, chronic inflammation, autoimmunity, neural hyperactivity, mast cell upregulation, accumulated oxidative stress, infection, urothelial lining defect, and extrabladder disorder were all proposed to be involved in the pathophysiology of IC/BPS [9].

Clinically, HIC/BPS is a unique inflammatory bladder disease characterized by urothelial denudation, which enhanced immune responses with lymphoplasmacytic infiltration, whereas NHIC/BPS was a noninflammatory disorder with little pathological changes in the bladder [22]. According to the International Society for the Study of Bladder Pain Syndrome (ESSIC) guideline, 53% of the IC/BPS patients displayed with detrusor mastocytosis (≥28 cells/mm^2^) and 50% with intrafascicular fibrosis [23]. Urothelial lining defect destroyed the permeability barrier and thereafter enhanced endothelial cell injury, which resulted in glomerulation hemorrhage after cystoscopic hydrodistention [24,25].

#### 3.2.1. Chronic Inflammation

Bladder biopsies from patients with IC/BPS shown increased numbers of mast cell, alterations in interstitial cells, infiltration of inflammatory cells, edema, fibrosis, and vascular lesions [26,27,28]. The inflammatory cells consisted of lymphocytes and plasma cells and distributed mostly in the suburothelial region. Lymphoid aggregates/follicles were observed in approximately 40% of IC/BPS patients [22]. Moreover, few eosinophils and neutrophils were shown in the bladder tissue. The superficial layer of urothelium is lost and urothelial denudation was often observed. Chronic inflammation cascade might damage to the glycosaminoglycan (GAG) layer of the urothelium and prompt extracellular matrix degradation in IC/BPS. Most of these alterations are consistent with chronic mucosa inflammation and are associated with voiding symptoms and pain in IC/BPS.

#### 3.2.2. Autoimmune Disorders

Autoimmune disorders were common accompanied with IC/BPS. Autoantibodies can be detected in the serum or urine of IC/BPS patients [5]. Autoimmunity against uroplakin induced suburothelial inflammation, serum antibody response and voiding dysfunction were observed in a murine cystitis model [29]. Increased mast cells in the bladder urothelium are seen in patients with IC/BPS [30]. Mast cells release several proinflammatory cytokines and chemokines as well as inflammatory mediators, such as neuropeptide substance *p* and nerve growth factor (NGF), which are associated with the proliferation of nerve fibers in IC/BPS [31,32,33,34]. In addition, bladder urothelium could release several neural signaling molecules, such as adenosine triphosphate (ATP) and NGF, and thereafter stimulate submucosal afferent nerves and activate mast cells [21], resulting in voiding and bladder pain symptoms and structural changes of bladder wall [35].

Tumor necrosis factor-α (TNF-α), a proinflammatory cytokine, in high concentration could cause excessive inflammation and bladder damage [36]. The TNF-α levels of serum [37] and bladder tissue [38] were significantly increased in IC/BPS patients, particularly in HIC/BPS. Moreover, IC/BPS patients also had elevated TNF-α level both in urine and bladder tissues [39]. Mast cell activation with the release of TNF-α elicited an inflammatory response in IC/BPS [40]. In an autoimmune IC model, interruption of mast cell activation by TNF-α inhibitor could improve bladder inflammation [41]. Treatment with adalimumab, an anti-TNF-α agent, significantly improved clinical symptoms in IC/BPS patients [42]. Certolizumab pegol is a monoclonal antibody specific to TNF-α. The certolizumab pegol treatment inhibited mast cell degranulation to release inflammatory mediators, such as histamine, prostaglandin, leukotriene, serotonin, heparin, and serine protease [43].

#### 3.2.3. Neurogenic Hyperactivity (Hyperexcitability)

Bladder inflammation might induce afferent nerve hyperactivity (hyperexcitability) and result in pain symptoms in IC/BPS [44]. For example, Toll-like receptor-4 (TLR-4) has been shown as important factor in central pain sensitization. Interactions between TLR-4-mediated inflammation and sex hormones were considered to be a potential mechanism in the different prevalence of pain condition in female and male IC/BPS patients [45]. Therefore, TLR-4 mediated inflammation was associated with painful symptoms and nerve hyperactivity (hyperexcitability), such as bladder pain, frequency, and urgency, especially in female IC/BPS patients.

Bladder afferent nerves are classified into two types: myelinated Aδ fibers and unmyelinated C fibers. Aδ fibers are thought to detect bladder filling under normal conditions, whereas C fibers are activated under pathological conditions. Bladder distension activated non-nociceptive Aδ afferent and triggered the normal sensation of bladder filling; while pathological conditions activated nociceptive C-fiber afferents leading to urinary urgency, increased voiding frequency, nocturia, urinary incontinence, and pain [46]. In IC/BPS patients, elevated sensory afferent activity was associated with C-fibers sensitization and caused bladder pain [47,48]. Mukerjiet al. revealed that the density of M2 and M3 receptors in the lamina propria was increased in the bladders of IC/BPS patients. There was a correlation between suburothelial muscarinic receptor density and urgency symptom scores [49]. The bladder tissue of HIC/BPS showed elevated levels of NGF, transient receptor potential vanilloid (TRPV) channels, ATP, and prostaglandins [50,51,52]. Overexpression of urothelial TRPV1 [53] and P2 × 3 receptors [54] and hypersensitivity of C-fiber pathway [55] are associated with urgency and detrusor overactivity.

The TRP superfamily was involved in the transduction of mechanosensory and nociception in LUT. The TRPV family consisted of four groups: TRPV1, TRPV2, TRPV4, TRPM4, TRPM8, and TRPA1 [56]. TRPV1 can be activated by vanilloids (capsaicin and resiniferatoxin (RTX)) involved in voiding function and pain sensation. TRPV1 receptors are essential for activating purinergic signaling in IC-induced bladder hyperactivity. Activation of urothelial cells with capsaicin or RTX increases intracellular calcium, leading to the induced release of NO and ATP, and eventually eliciting transient currents. NGF may participate in the pathogenesis of OAB syndrome through TRPV1 signaling [57]. TRPV1 played an important role in the symptoms of inflammatory pain, frequency, and urinary urgency in IC/BPS [58]. After botulinum toxin A (OnabotulinumtoxinA; BoNT-A) treatment, the number of suburothelial afferents expressing TRPV1 was significantly reduced [59].

TRPV4 is a Ca^2+^-permeable stretch-activated channel involved in stretch-induced ATP release to participate in bladder filling sensory pathways. Activation of urothelial-TRPV4 facilitated bladder reflexes via activation of mechanosensitive capsaicin-C fiber [60]. In the urinary bladder, TRPV4 is not only abundantly expressed in the urothelium but also localized in subepithelium, afferent neurons, and detrusor smooth muscles. Under physiological conditions, urothelium stretch caused a TRPV4-mediated Ca^2+^ influx into the cell, which triggers ATP release, and thus modulates afferent nerve activity in response to bladder filling during the urination cycle. TRPV4^−/−^ mice exhibited abnormal voiding frequency, increased frequency of nonvoiding contraction, augmented bladder capacity, and reduced ATP response to urothelial stretch [61]. In rat model with CYP-induced cystititis, HC-067047, a potent and selective TRPV4 antagonist reduced micturition frequency and increased functional bladder capacity [62].

#### 3.2.4. Urothelial Defect

The apical surface of the urothelium is coated with a layer of GAG, which included glycoproteins, proteoglycans, and glycolipids. Bladder urothelial GAG layer covers the umbrella cells in the superficial urothelial layer. The histopathological feature in IC/BPS was denudation or thinning finding of the bladder epithelium. Disrupted urothelium and urothelial barrier defects in IC/BPS resulted in diffusion of urine toxins, leading to bladder inflammation, detrusor interstitial fibrosis, and afferent nerve hyperactivity (hyperexcitability). The inflammatory response caused painful sensation and urinary storage symptoms in IC/BPS patients [22,35,63,64]. Compared to the control bladder tissue, the bladder tissue of IC/BPS patients had significantly decreased expression of tight junction proteins (e.g., E-cadherin, zonula occludens-1 (ZO-1)), impaired cell adhesion, alleviated cell proliferation in the basal layers, increased urothelial apoptosis, and strengthened oxidative stress protein [65,66,67]. Loss of GAG layer was associated with a loss of biglycan and perlecan on the luminal layer [68]. Denudation or anatomical loss of urothelium consistency was reported in HIC/BPS patients [22,63]. Intravesical therapy with chondroitin sulfate and GAG substitutes for IC/BPS patients was aimed to reconstitute the integrity of the epithelium through the binding of GAGs to proteoglycans with structural urothelium [69]. Although GAGs in the bladder urothelium have an important role, further studies to identify the key molecules in IC/BPS will help to improve the efficacy of treatment and identify biomarkers of the disease.

#### 3.2.5. Oxidative Stress: Nrf2-ARE Signaling Pathway

The cellular antioxidative response transcription factor, Nrf2 (nuclear factor E2-related factor 2), is bound with Kelch-like ECH-associated protein 1 (Keap1) in the homeostatic conditions. Nrf2 dissociates from Keap1 and translocates from cytoplasm into the nucleus under oxidative stress. The nucleus Nrf2 initiates the expression of a series of antioxidant gene (e.g., SOD, glutathione reductase, and heme oxygenase-1 (HO-1)) [70,71,72]. The Keap1-Nrf2 stress response pathway is the inducible protective response against oxidative stress by regulating the expression of cytoprotective genes. Under homeostatic conditions, Keap1 forms part of an E3 ubiquitin ligase that regulates Nrf2 expression through ubiquitination and proteasome degradation. However, in response to stimulation by excessive oxidative stress, Keap1 assists Nrf2 to get away from cellular ubiquitination via cysteine oxidation. Nrf2 then translocates into the nucleus and binds to AREs to promote the expression of downstream genes, including phase II detoxifying enzymes (antioxidant enzymes (SOD, CAT, and GSH-Px, glutamate–cysteine ligase (GCL)), NADPH, and HO-1) to inhibit the production of oxidative stress [73,74].

Previous study revealed that oxidative stress is closely associated with pathological mechanisms of IC/BPS [75]. In CYP-induced IC animal model, Ni et al. indicated that CYP treatment could increase urinary frequency and urgency, pain sensitization, decreased contractility, bladder edema, oxidative stress disorder, and the mRNA and protein levels of antioxidant genes downstream of Nrf2 pathway. The Nrf2^−^^/^^−^ CYP mice had more severe symptoms but no significant changes in the mRNA and protein levels of antioxidant genes downstream of Nrf2 pathway. Therefore, upregulating antioxidant genes and inhibiting oxidative stress by Nrf2 may protect from bladder injury and ameliorate bladder dysfunction in IC/BPS [76]. The Nrf2-antioxidant response element signaling pathway controls the translational expression of genes involved in the detoxication and elimination of reactive oxidants by promoting antioxidant capacity to activate cellular defense against oxidative stress [77]. For example, elevated Nrf2 expression in the peripheral blood leukocytes was observed, while downregulation of the Keap1 was found in IC/BPS.

#### 3.2.6. Abnormal Angiogenesis

Angiogenesis played a vital role in maintaining blood vessels delivering nutrition and oxygen to supply the regeneration of dysfunctional bladder. The mechanism of angiogenesis by vascular endothelial growth factor (VEGF) signaling pathway is through the VEGF receptor and involves the stimulation of phosphorylation of Erk1/2, P38, and Akt [78]. Elevated VEGF levels increased angiogenesis in HIC/BPS [5,79,80,81,82]. Abnormal angiogenesis in bladder tissues is closely related to urinary frequency and bladder pain in patients with IC/BPS [79,81]. Increased and dysregulated angiogenesis is also implicated with mucosal bleeding after distension in NHIC/BPS [79]. Increased expression of hypoxia-inducible factor-1α (HIF-1α) and VEGF is associated with glomerulation formation in patients with IC/BPS [79]. The expression of tissue necrotic factor-α (TNF-α), VEGF, CD31 and transforming growth factor-ß (TGF-ß) was significantly increased in IC/BPS patients. In contrast, a significant increase in the expression of mast cell, tryptase, and collagen was observed in HIC/BPS patients. Increased VEGF is associated with bladder inflammation in patients with IC/BPS [83]. These results suggest that bladder angiogenesis was correlated with urinary frequency and bladder pain in IC/BPS patients.

#### 3.2.7. Exogenous Urine Substances

Substances in urine might act as toxic or harmful irritants. Damage-associated molecular patterns from the degenerated cells by toxic substances might promote immune response and pathological inflammation. For example, urinary metabolites of ketamine are known to induce inflammation of the bladder associated with immune hypersensitivity [84]. These symptoms of ketamine-induced ulcerative cystitis (KIC) are similar to those of IC/BPS. Elevated serum IgE may be associated with development of KIC. In ketamine addiction patients, pathological features observed by endoscopy include bladder erythematous mucosa, mucosa ulceration and laceration, wall thickening, hydronephrosis, and ureter mucosa swelling [85,86]. Clinically, patients with KIC have increased bladder eosinophil cell and mast cell infiltration with elevated serum immunoglobulin-E (IgE) levels associated with hypersensitivity and/or allergic reactions [84,87,88]. The pathological changes in the animal model of KIC were denuded urothelium, neurogenic inflammation, abnormal apoptosis, bladder wall thickening, and infiltration of mast cells, eosinophils, lymphocytes, and plasma cells [88]. Previous evidence suggested that the toxic effect of ketamine metabolites results in bladder barrier dysfunction, neurogenic inflammation, IgE-mediated inflammation, and nitric oxide synthase-mediated inflammation, all of which contribute to the etiology of KIC [88].

## 4. Histopathology

### 4.1. Histopathological Evaluation of Bladder Biopsy

Histological differences between HIC/BPS and NHIC/BPS are shown in Table 1. HIC/BPS is associated with severe inflammation of the urinary bladder accompanied by lymphoplasmacytic infiltration and urothelial denudation, whereas painful bladder syndrome has little pathological changes in the bladder. Clinically, HIC/BPS was associated with severe inflammation of the urinary bladder accompanied by lymphoplasmacytic infiltration and urothelial denudation, whereas NHIC/BPS showed little pathological changes in the bladder [22]. According to the International Society for the Study of Bladder Pain Syndrome (ESSIC) guideline for IC/BPS, 53% of the patients present with detrusor mastocytosis (≥28 cells/mm^2^) and 50% with intrafascicular fibrosis [23]. Urothelial defect destroyed the permeability barrier and endothelial cell injury, resulted in glomerulation hemorrhage after cystoscopic hydrodistention in IC/BPS bladders [24,25].

### 4.2. Infiltration of Lymphocytes and Plasma Cells

The histopathology of IC/BPS was found to increase stromal fibrosis and mast cell counts, which may induce local inflammation to limit bladder distention [89], leading to a small functional bladder capacity and symptoms of urination frequency and urgency. Inflammatory cell infiltration is observed in the suburothelial region in IC/BPS patients. Lymphoid follicles are frequently present. For types of infiltrating inflammatory cells in HIC/BPS, lymphocytes and plasma cells are dominant, while plasma cells are few in NHIC/BPS.

### 4.3. Mast Cell Infiltration and Neurogenic Inflammation

The role of mast cells might be implicated differently between ulcerative subtype and nonulcerative subtype IC/BPS [90,91]. Increased stromal fibrosis and mast cell counts were observed in bladder of IC/BPS without Hunner lesion [92]. Mast cells play a pivotal role in the pathogenesis of HIC/BPS. The role of mast cells in IC/BPS pathogenesis are implicated in systemic disorders with afferent hypersensitivity and neurogenic inflammation [93].

## 5. Histopathological Differences between OAB and IC/BPS

Several studies have linked OAB and IC/BPS to chronic inflammation, showing that the levels of bladder and urinary NGF, cytokines, and serum C-reactive protein are elevated in both patients with OAB and those with IC/BPS [52,68,94,95,96,97,98]. The expression of E-cadherin and ZO-1 was decreased in IC/BPS, but not in OAB patients, suggesting a prominent barrier function of urothelium in IC/BPS but not altered in the OAB bladders [66]. OAB and IC/BPS might share a common pathway, although mast cell infiltration was found in both diseases, while abnormal urothelial barrier function only occurred in patients with IC/BPS, and not in those with OAB [66]. Differences between IC/BPS and OAB are shown in Table 2.

## 6. Clinical Diagnosis for IC/BPS

Urinalysis for evaluation for IC/BPS patients usually has no abnormality. The 3-day urinary diary showed increased urinary frequency and declined voided volumes [99]. High signal intensity of the bladder wall in diffusion-weighted magnetic resonance imaging (MRI) had been reported in IC/BPS [100].

### 6.1. Cystoscopy

In cystoscopy of patients with IC/BPS, the most common finding is glomerulation hemorrhages. In cystoscopy, IC/BPS is diagnosed when the bladder has been filled to its maximum capacity (at a pressure of 80–100 cm H_2_O). In IC/BPS patients, mucosal splitting, glomerulations, and Hunner ulcers are frequently observed mucosal damage in IC/BPS [101]. In order to diagnosis of the HIC/BPS or NHIC/BPS, cystoscopy is recommended to examine the bladder mucosa after bladder filling and determine the presence or absence of Hunner lesions [102,103]. Cystoscopy for Hunner’s disease requires fulguration or resection of lesions concomitantly with hydrodistension to improve treatment outcome. The presence or absence of Hunner ulcer in IC/BPS patients is believed to have an important role in symptom variations, differences in therapeutic success, and the level of pain, especially the pain related to bladder distension [104,105].

### 6.2. Bladder Capacity

Evaluated mucosal gene expression in bladder biopsies from IC/BPS patients found a clear segregation of expression profiles based on a low (≤400 cc) versus a nonlow (>400 cc) anesthetic bladder capacity [106]. The low bladder capacity group was found to have increased expression of genes involved in inflammation and the immune response as well as decreased expression of genes important for bladder mucosal barrier integrity. These molecular and clinical data supported the framework for differing phenotypes of IC/BPS: a low bladder capacity subtype with bladder-centric disease and a nonlow bladder capacity subtype with generalized pain and psychosomatic disease. Furthermore, previous studies have shown that IC/BPS patients with low bladder capacity were older and had higher levels of frequency and nocturia, higher levels of acute and chronic bladder inflammation, earlier onset of painful urge during bladder filling, and poorer compliance [107,108].

### 6.3. Symptomscore

There are several questionnaires used to evaluate the severity of IC/BPS, including O’Leary–Sant Interstitial Cystitis Symptom and Problem Index (Interstitial Cystitis Symptom Index (ICSI) and Interstitial Cystitis Problem Index (ICPI)) [109], visual analog scale pain scores (VAS), the Genitourinary Pain Index (GUPI), the Wisconsin Interstitial Cystitis Scale (UW-IC Scale) [110], and the Pain, Urgency, and Frequency (PUF) score [111]. Both ICSI and ICPI scores are the most frequently used questionnaires to diagnose and evaluate the therapeutic outcome in patients with IC/BPS [112]. Moreover, OSS and VAS scores are also useful in IC/BPS patients with or without Hunner lesions.

### 6.4. Diagnostic Hydrodistension

The bladder usually looked normal before distension but might undergo mucosal bleeding after hydrodistension to maximal bladder capacity [5]. Mucosal bleeding or glomerulations hemorrhage after distension and Hunner ulcers might represent different bladder pathology, although convincing evidence as a definite diagnostic criterion is lacking [82,113]. The literature reported that glomerulations should be included in the diagnosis or phenotyping of IC/BPS. However, there is no consistent relationship between glomerulations and the diagnosis of IC/BPS [113].

### 6.5. Serum and Urine Specimens

The urine of IC/PBS patients resides in the bladder for a long time to capture proteins and peptides shed from the bladder. Abnormal expressions of several serum and urine specimens, including macrophage inflammatory protein, growth factor, histamine, glycoprotein, and cytokines may serve as a disease-specific biomarkers in the diagnostic or prognostic tool for the clinical management in IC/BPS. There is a clear preference for noninvasive biomarkers, such as proteins found in urine and serum. In addition, compared with the serum proteome, the urine proteome showed a better association with IC/PBS symptoms.

### 6.6. Bladder Biopsy

Bladder biopsies from IC/BPS patients showed that there were macrophages and eosinophils in the urothelium and an increased number of mast cell count in the detrusor [24,35]. According to histopathological evaluation of bladder biopsy, increased microvessels, edema, fibrosis, hemorrhage, and fibrin exudation are found in variable degrees, which should be the result of chronic inflammation and epithelial denudation [82]. In biopsies collected from IC/BPS patients, the pathohistology showed epithelial denudation, increased density of suburothelial afferent nerve fibers, and infiltration of inflammatory cells such as mast cells [22,114,115,116,117]. These findings would be helpful for the definite diagnosis of the HIC/BPS or NHIC/BPS. Aberrant expressions of several bladder urothelial markers, including mast cells, epithelial differentiation proteins, cell membrane proteins, neurotransmitters, and cytokines, were observed in IC/BPS.

## 7. Current Investigated Biomarkers in IC/BPS

In recent years, the black box representing BPS/IC has expanded from the bladder-centered concept to the whole body [118]. The pathophysiology of IC/BPS is composed of multiple factors. In order to avoid the risk of invasive cystoscopic procedures, development of noninvasive biomarkers to discriminate HIC/BPS from NHIC/PBS is of high priority [106,119]. Biomarkers currently used in diagnosis, therapeutic efficacy, determination of treatment safety, and advancing the mechanistic understanding of IC/BPS patients are summarized in Table 3. Identifying the key molecules by using adequate sample size and selecting controls for in IC/BPS will help to improve the efficacy of treatment and identify biomarkers of the disease [118].

### 7.1. Urothelial Associated Proteins

Proper function of the urothelium requires normal epithelial integrity, which relies on intercellular adhesion molecules and a layer of molecular components on the apical surface of the urothelium, which is composed of GAG. Abnormal expressions of urothelial-associated proteins, including zonula occludens type 1 (ZO-1), E-cadherin, uroplakin, chondroitin sulfate, and receptors/ion channels have been noted in IC/BPS bladders [68,120,121]. For example, E-cadherin is one of the intercellular junction proteins that have been suggested to be involved in the barrier function of the urothelium. The role of E-cadherin in the pathophysiology of IC/BPS remains controversial. Recent studies revealed decreased or abnormal expression of E-cadherin was related to increased bladder permeability in IC/BPS [65,66,68]. E-cadherin expression was significantly decreased in HIC/BPS patients compared to NHIC/BPS patients. Uroplakins are a family of integral membrane proteins of bladder urothelium. Overexpression of uroplakin III has also been shown in bladder of NHIC/BPS [121]. In an animal model of experimental autoimmune cystitis, injection of UPK3A has been shown to induce T-cell attack on the bladder epithelium, resulting in chronic suprapubic hypersensitivity and other symptoms that mimic human IC/PBS disease [122]. These abnormal alterations may help disrupt urethral barrier and sensory functions, leading to increased afferent nerve activity and manifesting bladder symptoms such as hypersensitivity, pain, or urgency.

### 7.2. Proinflammatory Cytokines or Chemokines

Several cytokines and chemokines were identified to be associated with IC/BPS and can serve as useful tools to assess treatment outcome. Overexpression of some proinflammatory genes has also been found in IC/BPS bladder [38,50,106]. Patients with HIC/BPS show increased expression of T- and B-cell markers in the submucosa [123]. Immunological reaction occurred in IC/BPS patient bladder had elevated level of serum IgE [124]. Meanwhile, increased levels of cytokine and chemokine have been found in the urine of IC/BPS patients. Erickson et al. [125] and Sakthivel et al. [126] found that several proinflammatory mediators, such as interleukin-6 (IL-6) and CXC chemokines, were increased in both urinary and serum samples of IC/BPS patients. Lamale et al. proposed the use of a combination of methylhistamine and IL-6 as a sensitive and specific marker for IC/BPS [127]. Ogawa et al. also confirmed that the mRNA of several CXCR3-binding chemokines (CXCL-9, 10, and 11) in patients with HIC/BPS [38] were elevated. The serum levels of IL-1ß, 6, 8, and TNF-α were significantly higher in the serum of IC/BPS patients than in control patients [128,129,130]. Several studies have linked OAB and IC/BPS to chronic inflammation, showing that the levels of bladder and urinary NGF, cytokines, and serum CRP are elevated not only in OAB patients but also in IC/BPS patients [52,68,94,95,96,97,98]. Both OAB and IC/BPS might share a common pathway, for example, mast cell infiltration was found in both diseases. However, abnormal urothelial barrier function only occurred in IC/BPS patients, but not in those with OAB [66]. Urine CXCL-10 is elevated in patients with IC/BPS, but not in OAB patients [131]. The upregulated levels of serum TNF-α, IL-1β, 6, and 8, and urine CXCL-10 level in IC/BPS patients might help provide as an appropriate diagnostic tool. The increased expression of proinflammatory cytokines and chemokine levels in the serum of IC/BPS patients indicated that not only the activation of mast cell, but also inflammatory mediators might play key roles in the pathogenesis of IC/BPS [132]. Serum CRP is elevated in patients with LUTS and IC/BPS [94]. Therefore, CRP might be useful as a biomarker for monitoring disease conditions and response to therapeutic interventions in LUTS patients. The CRP levels of serum and urine might serve as a biomarker of local bladder inflammation to distinguish patients with IC/BPS.

### 7.3. Growth Factor

Potential biomarkers associated with IC/BPS include NGF, vascular endothelial growth factor (VEGF), epidermal growth factor (EGF), and heparin-binding epidermal growth factor-like growth factor (HP-EGF) [133]. Analysis of urinary and serum biomarkers in IC/BPS patients may provide insight into the development and treatment outcome of the disease. Studies suggested that growth factor markers may help to differentiate between HIC/BPS and NHIC/BPS patients.

Serum NGF levels have been found to increase in several systemic diseases, including allergic disease, autoimmune disease, psychosocial stress and lower urinary tract disease [134,135,136]. Patients with lower urinary tract diseases, including stones, tumors, acute bacterial infection, IC/BPS [52,97,98,137,138], and bladder outlet obstruction [139], have been found to increase NGF levels in the urine, serum, and/or bladder tissue. In the bladder, NGF is expressed in the urothelium, smooth muscle, afferent nerves, and ganglia [140]. NGF acts as a chemical mediator in C-fiber afferents that may regulate urinary bladder function [141,142]. Current findings suggested that the urinary NGF level can be monitored as a biomarker for IC/PBS severity and for treatment response. In a transgenic mouse model, NGF overexpression in the bladder led to neuronal hypersensitivity and changed in urinary bladder function [143]. In samples of patients with IC/BPS, increased levels of NGF have been noted in the urine [52,97,137,138] and bladder tissue [98]. The NGF level of serum and urinary in IC/BPS patients was elevated, while the level was also not related to the severity of IC/BPS [144]. The urinary NGF level has been shown to be closely related to the visual analog scale (VAS) score for inflammatory pain and treatment outcome for IC/BPS. Clinical and experimental data in IC/BPS have indicated correlation between increased levels of NGF in the bladder tissue and urine and painful inflammatory conditions. These findings suggested that NGF is associated with bladder function, and elevated urinary NGF levels reflect that chronic inflammation occurs in the urinary bladder of IC/BPS patients. NGF might be developed as an indicator for treatment, in order to be a sensitive molecular diagnostic tool for IC/BPS.

Ischemia/hypoxia of the bladder mucosa leads to IC/BPS symptoms. VEGF is a signal protein that stimulates the formation of blood vessels to restore oxygen supply to tissues under hypoxic conditions [79]. Bladder urothelium from IC/BPS patients has been shown to exhibit higher VEGF level to induce bladder fibrosis and reduce bladder capacity [79]. The increased expression of CD31 in bladder tissues was correlated with O’Leary–Sant problem indexes and VAS scores [81]. Intravesical instillation of VEGF-modulated sensory and motor nerve plasticity improved bladder function and visceral sensitivity in rats [145]. In addition, compared with the control group, systemic anti-VEGF neutralizing antibody pretreatment significantly reduced the rat’s pelvic pain response to CYP-induced IC [146]. Urinary symptom with pain severity was significantly correlated with urinary VEGF levels in female IC/BPS patients [147]. Bladder urothelium of IC/BPS patients exhibited meaningfully higher expressions of HIF-1α and VEGF, induced bladder fibrosis, and reduced bladder capacity after chronic inflammation. Furthermore, VEGF expression level was associated with bladder pain severity and glomerulation [79,148]. The role of VEGF was the crucial urine markers to discriminate IC/BPS patients from OAB patients [149].

In summary, these findings suggested that the increased levels of VEGF in bladder tissues or urine could be associated with angiogenesis and provide new insights into the pathophysiological basis of IC/BPS [81]. Some inflammatory proteins were associated with increased angiogenesis and glomerulation in IC/BPS, including VEGF [79] and hypoxia-inducible factor 1-α (HIF-α) [150]. In patients with IC/BPS, increased urothelial VEGF expression was associated with bladder inflammation and bladder capacity reduction. The expression level of VEGF in IC/BPS bladder tissue declined after repeated BoNT-A injection [83]. Therefore, intravesical BoNT-A injection reduced the expression of VEGF associated with a concomitant decrease in inflammatory marker levels in patients with IC/BPS [83,151].

### 7.4. Mast Cells and Histamine

The roles of histamine and histamine receptors in mast-cell-mediated allergy and inflammation have been documented in IC/BPS. Several reports have indicated that bladder specimens of patients with IC/BPS show increased numbers of infiltrated mast cells (mastocytosis) [152,153,154,155]. Other studies have also shown that mast cells are found both in the epithelium and in bladder washings of patients with IC/BPS, but not in normal individuals [154,156]. Mastocytosis in IC/BPS is best documented by tryptase immunocytochemical staining. Standard surgical stains such as Giemsa and toluidine blue routinely underestimate the degree of mastocytosis. Mast cells are six- to eightfold higher in the detrusor in HIC/BPS group and two- to threefold higher in NHIC/BPS group compared with control group. Detrusor mastocytosis occurs in both HIC/BPS and NHIC/BPS. Mucosal mast cell increase is present in NHIC/BPS. Mast cell activation occurs in the mucosa and submucosa, but without typical exocytosis. Mast cell activation, regardless of bladder location or degree of mastocytosis, is important. Mast cell-derived vasoactive and proinflammatory molecules may contribute to the pathogenesis of IC/BPS. This evidence suggests that IC/BPS is mediated by the immune system, and the abnormalities are possibly caused by dysregulation of the inflammatory response. In addition to the increased number of mast cells in the bladder in IC/BPS, the mast cells are mainly activated, as they are partially or completely degranulated [153]. Thus, the increased cell number and the activation of mast cells in the bladder suggested that mucosal mast cells are involved in the etiology of IC/BPS and the overexpression of mast cells as a biomarker of IC/BPS.

### 7.5. C-Reactive Protein (CRP)

CRP is secreted by the liver in response to inflammatory processes. Serum CRP level might be used to differentiate IC/BPS patients from those with bladder hypersensitivity disorders. The NGF levels of urine and bladder tissue as well as the cytokines and C-reactive protein (CRP) levels of serum were increased in OAB and IC/BPS [52,94]. CRP is a common biomarker of inflammation and infection for heart diseases, and serum CRP level is used to determine disease progression or treatment effectiveness. An elevation of CRP in the bladder tissue and urine has been associated with chronic inflammation and LUTs [94]. Serum CRP is elevated in patients with LUTS and IC/BPS [94,157]. Therefore, CRP might be useful as a biomarker for monitoring disease conditions and response to therapeutic interventions in LUTS patients. The CRP levels of serum and urine might serve as a biomarker of local bladder inflammation to distinguish patients with IC/BPS.

### 7.6. ATP

ATP is released from urothelium in response to bladder stretch and could act on urothelial purinergic receptors. Patients with IC/BPS have increased afferent nerve density and ATP release, which might affect the symptoms of pain, urgency and frequency [101]. The expression of both P2X and P2Y receptors in nerve fibers and myofibroblasts, located close to urothelium and detrusor muscle, and the sensitivity of these receptors to ATP suggest that ATP release may influence function of myofibroblasts and afferent nerve endings [158]. In patients with IC/BPS, urinary ATP levels were significantly higher than control [159]. Blocking ATP release improved the symptoms of pain, urgency, and frequency for IC/BPS patients. Similar to the data in human IC/BPS, a significant increase in stretch-evoked ATP release in IC/BPS feline model [160] and in CYP-induced rats caused chronic bladder inflammation [161].

In addition, inhibition of purinergic P2X3 receptors on afferent terminals resulted in decreased ATP release from the urothelium and improved the painful sensations in IC/BPS. Clinically, inhibition of efferent ATP release treated with BoNT-A could ameliorate acute pain and urgency sensation [162]. Purinergic receptor antagonists show positive results in the treatment of different symptoms of IC/BPS [101]. In IC/BPS patients, elevation of urinary ATP level and increase stretch-activated ATP released by bladder urothelium has been reported, suggesting augmented purinergic signaling in IC/BPS bladders [163]. Though ATP and purinergic receptors may play an important role in modulating bladder function, the mechanisms underlying activation of the micturition pathway at lower bladder volumes and mediators involved are not fully understood.

### 7.7. Antiproliferative Factor (APF)

APF glycoprotein is secreted by bladder urothelial cells from IC/BPS patients and slows down the growth of urothelial cells [164,165,166]. APF may mediate the pathological changes observed in IC/BPS, including inhibition of cell growth, increased barrier permeability and reduced proteins expression (e.g., cadherins) [65], while promoting the formation of intercellular complexes. Increased susceptibility to urothelial damage may be due to altered factors that regulate the development of structural elements. Therefore, these proteases have been proposed as potential biomarkers or to provide assessment of disease progression but have not been validated in lower urinary tract disorders. An increased APF and lower expression of IL-8 have been found in IC/BPS bladders, which may contribute to IC/BPS pathophysiology [167,168,169].

### 7.8. Cyclooxygenase-2 (COX-2) and Prostaglandin E_2_ (PGE_2_)

PGE_2_ production is initiated by activation of PLA_2_, which releases arachidonic acid from membrane phospholipids and COX. The urothelial cells produce several prostaglandins including PGE_2_. COX-2 is an inducible enzyme responsible for the production of prostaglandins, including PGE_2_, at the site of the inflammation. Inhibition of COX-2 overexpression was related to hemorrhagic cystitis [170]. The COX-2/PGE_2_ pathway has been involved in chronic inflammation. Previous study showed the association of inflammation with OAB symptoms by the significant elevation of urinary PGE_2_ level in OAB patients [171]. Studies revealed that urine levels of PGE_2_ were increased in the HIC/BPS patients [51]. The increasing expression of PGE_2_ via COX-2 upregulation in the bladder may be activating afferent nerves and contributing to bladder hypersensitivity and pain in IC/BPS.

### 7.9. Methylhistamine

Stimulation of mast cells has been shown to promote the degranulation and release of vasoactive, proinflammatory, and nociceptive mediators in bladder tissue, including histamine, cytokines, and proteolytic enzymes [172]. Methylhistamine, known as histamine metabolite, was measured using radioimmunoassay kits and was normalized to urinary creatinine levels [127]. Monocyte chemoattractant protein-1 (MCP-1) upregulated in IC/BPS was a possible contributing factor for inducing mast cell degranulation and releasing histamine from mast cells. Histamine released from mast cells plays a key role in neural sensitization that is responsible for IC/BPS-related bladder and urinary pain [173]. Therefore, histamine levels have been used as a biomarker for IC/BPS in genetic studies [127].

### 7.10. GP51

The pathophysiology of IC/BPS urothelium is involved in an aberrant synthesis of bacterial defense molecules such as GP51 [174]. The level of urinary glycoprotein GP51 secreted from urothelial cells was reduced in IC/BPS patients [174]. The urinary glycoprotein GP51 might serve as a clinical marker for interstitial cystitis [174].

Taken together, the potential biomarkers of urothelial barrier protein (Uroplakin III, E-Cadherin, and ZO-1), apoptotic signaling molecules (Bad, Bax, and Cleaved caspase-3), HIF-1α, and TRPV1, 2, and 4 need to be identified from bladder biopsy and further analysis using real-time PCR for RNA expression and using Western blot or immunohistochemistry stain for protein expression. The other potential biomarkers of proinflammatory cytokines, chemokines, and proteins (CXCL-1, CXCL-9, CXCL-10, CXCL-11, IL-1β, IL-2, IL-4, IL-6, IL-8, TNF-α, and IgE), growth factors (NGF, VEGF, HB-EGF, EGF, and APF), GP51, ATP, CRP, methylhistamine, PGE2, and platelet-derived endothelial cell growth factor/thymidine phosphorylase (PDECGF/TP) can be analysis from urine supernatant samples and serum samples and further analysis by enzyme-linked immunosorbent assay for suspended protein expression, which is more rapid, popular, easy, and noninvasive than bladder biopsy analysis. In addition, the urine proteome showed a better association with IC/PBS symptoms than the serum proteome.

**Table 3 diagnostics-12-00075-t003:** Potential biomarkers of bladder tissue, urine, and serum for the diagnosis of IC/BPS.

Biomarkers	Species	Sample	Changes	References
Urothelial barrier protein				
Uroplakin III	Human	Urine	Elevated	[121]
	Human	Bladder tissue	Decreased	[121]
E-Cadherin	Human	Bladder tissue	Decreased	[65,66,68,87]
ZO-1	Human	Bladder tissue	Decreased	[65,66,68,87]
Apoptotic signaling molecules				
Bad	Human	Bladder tissue	Elevated	[175]
Bax	Human	Bladder tissue	Elevated	[175]
Cleaved caspase-3	Human	Bladder tissue	Elevated	[175]
Proinflammatory cytokines, chemokines and proteins				
CXCL-1	Human	Urine	Elevated	[119]
CXCL-9	Human	Bladder tissue	Elevated	[37,38]
CXCL-10	Human	Serum	Elevated	[37,38]
	Human	Bladder tissue	Elevated	[38]
	Human	Urine	Elevated	[119,131]
CXCL-11	Human	Bladder tissue	Elevated	[37,38]
IL-1β	Human	Serum	Elevated	[37]
IL-2	Human	Urine	Elevated	[130]
IL-4	Human	Urine	Elevated	[151,176]
IL-6	Human	Serum	Elevated	[37]
	Human	Urine	Elevated	[119,127,130]
IL-8	Human	Bladder tissue	Decreased	[169]
	Human	Serum	Elevated	[37]
	Human	Urine	Elevated	[130]
TNF-α	Human	Serum	Elevated	[37]
IgE	Human	Serum	Elevated	[124]
Growth factors				
NGF	Human	Bladder tissue	Elevated	[37,177,178]
	Human	Urine	Elevated	[52,119]
	Human	Serum	Elevated	[144]
VEGF	Human	Serum	Elevated	[83]
	Human	Bladder tissue	Elevated	[79,83]
HB-EGF	Human	Urine	Decreased	[179]
EGF	Human	Urine	Elevated	[179]
APF	Human	Urine	Elevated	[167,179]
Other potential biomarkers				
HIF-1α	Human	Bladder tissue	Elevated	[24,79]
GP51	Human	Urine	Decreased	[174]
ATP	Human	Urine	Elevated	[163]
CRP	Human	Serum	Elevated	[94]
TRPV1, 2, 4	Human	Bladder tissue	Elevated	[50]
PDECGF/TP	Human	Serum	Elevated	[37]
PGE_2_	Human	Urine	Elevated	[51]
Methylhistamine	Human	Urine	Elevated	[127]

Note: APF, antiproliferative factor; ATP, adenosine triphosphate; CRP, C-reactive protein; CXCL, CXC chemokine ligand; CXCR, CXC chemokine receptor; EGF, epidermal growth factor; FGF, fibroblastic growth factor; GP51, glycoprotein 51; HB-EGF, heparin-binding epidermal growth factor; HIF-1α, hypoxia-inducible factor-1-α; IC/BPS, interstitial cystitis/bladder pain syndrome; IL, interleukin; NGF, nerve growth factor; PDECGF/TP, platelet-derived endothelial cell growth factor/thymidine phosphorylase; PGE_2_, prostaglandin E_2_; TNF-α, tumor necrosis factor-α; TRPV, TRP vanilloid; VEGF, vascular endothelial growth factor; ZO-1, zonula occludens-1.

## 8. Impact of Potential Biomarkers for Assessment of IC/BPS Therapeutic Outcome

The impact of potential biomarkers for assessment of IC/BPS therapeutic outcome was discussed, including intravesical instillation with hyaluronic acid (HA), chondroitin sulfate (chondroitinsulphate), pentosan polysulfate (pentosanpolysulfate), and heparin sulfate, resiniferatoxin, dimethyl sulfoxide (DMSO), alkalinized lidocaine to restore GAG layer and intravesical injection with BoNT-A, low intensity extracorporeal shock wave therapy (LiESWT), regenerative medicine therapy with platelet-rich plasma (PRP), electrostimulation, acupuncture, endoscopic surgery, and medication. Several cytokines and mediators identified to be associated with the disease can be useful tools to assess treatment outcome. Classification of IC/BPS will be required to explore new biomarkers for diagnosis, as well as for treatment efficiency in IC/PBS patients. The potential biomarkers are summarized in Table 4.

### 8.1. Intravesical Instillation with GAG, Hyaluronic Acid (HA), Chondroitin Sulfate and/or Pentosan Polysulfate

The pathophysiology of IC/BPS urothelium involves abnormal differentiation, leading to altered synthesis of proteoglycans, tight junction proteins, cell adhesive proteins, and bacterial defense molecules (e.g., GP51) [174,180]. Therefore, replacement therapies have been effectively used to treat IC/BPS (e.g., intravesical glycosaminoglycan instillation) [181]. The main components of GAG glycoproteins in the bladder included HA, chondroitin sulfate, pentosan polysulfate, and heparan sulfate, which contribute to urothelial barrier functions against toxic substances, carcinogens, and microorganisms from the urine [182]. HA regulates cellular activities from migration, proliferation, differentiation, and inflammation to cell–cell interactions and cell–extracellular matrix adhesions. Some studies implicated that HA is involved in wound healing, tissue repair, and angiogenesis [183,184,185]. HA plays the role of a negative regulator in inflammatory activation and protector of cells against free radical damages. Bladder instillation of HA is one of GAG replacement therapy for IC/BPS, which improves bladder pain and storage symptoms (i.e., urgency, frequency, and nocturia) of IC/BPS [186,187,188,189,190,191,192,193,194]. Kim et al. found that 77.2% of patients with IC/BPS had improvement in bladder pain after three months of HA instillation, but about half of these patients showed persistent urinary frequency [195]. Jiang et al. reported that the level of urinary NGF, but not brain-derived neurotrophic factor (BDNF), decreased in IC/BPS patients treated with HA [196]. Long-term intravesical HA, chondroitin sulfate, and combination therapy for patients with IC/BPS all showed symptom improvement [197].

### 8.2. Intravesical Instillation or Injection with BoNT-A

BoNT-A can relieve pain, urgency, and frequency for refractory IC/BPS and improve the functional bladder capacity [198,199]. Kuo et al. suggested that functional bladder capacity (136 ± 79.1 vs. 208 ± 110, *p* = 0.000) was significant improvement after four repeated injections with BONT- A in IC/BPS patients. Nevertheless, BoNT-A has anti-inflammatory and antinociceptive effects in the treatment of OAB [200,201,202]. Several studies have shown that therapeutic application of BoNT-A reduced bladder pain in patients with refractory IC/BPS [203,204]. The increased NGF levels in the bladder tissue of IC/BPS patients decreased to the normal range after BoNT-A treatment [205]. In patients with detrusor overactivity treated with BoNT-A, the expressions of TRPV1 and P2X3 receptors were reduced on the suburothelial sensory afferents [202]. Therefore, BoNT-A therapy for neurogenic detrusor overactivity and refractory OAB have shown significant improvement of urinary urgency, bladder pain, and bladder capacity. However, invasive intravesical injection procedure may result in side effects, including hematuria, increased residual urine, nerve pain, acute urinary retention, and UTI [206]. Liu et al. also demonstrated that BoNT-A bladder injection can reduce the urine level of nerve growth factor, decrease neurogenic inflammation, and inhibit the action of proinflammatory cytokine, IL-1 [205]. In addition, BoNT-A can inhibit the transmission of TRPV1 to neuron cell membranes [207]. TRPV1 has been confirmed to play an important role in the symptoms of IC/BPS, such as urinary urgency, frequency, and inflammatory pain. Liu et al. showed that increased severity of bladder inflammation was associated with a higher expression of TRPV1-immunoreactive nerve fibers in IC/BPS [58]. Recently, intravesical injection of BoNT-A was shown to exhibit anti-inflammatory effects for IC/BPS [186,208,209,210]. In addition, inhibition of purinergic P2X3 receptors on afferent terminals, leading to a decrease in ATP release from the urothelium, can improve the painful sensations in IC/BPS [58]. A previous study showed that intravesical injections of BoNT-A can reduce bladder pain in patients with refractory IC/BPS. The average effective duration is around 6 months [211]. NGF levels in the bladder tissue are significantly increased in patients with IC/BPS, and they decreased to the normal range after BoNT-A treatment [205]. The summary of therapeutic efficacy assessments for intravesical instillation or detrusor muscle injection with BoNT-A for IC/BPS is shown in Table 4 and Figure 1.

### 8.3. Low Intensity Extracorporeal Shock Wave Therapy (LiESWT)

LiESWT has been used widely for various types of urological diseases, including erectile dysfunction (ED), OAB, CPPS, and IC/BPS. LiESWT improved bladder function by promoting angiogenesis [212], inhibited the production of ROS and ameliorated oxidative stress in rats with CYP-induced acute cystitis [213]. LiESWT treatment inhibited NGF, IL-6, and COX-2 expression, and blocked bladder pain, inflammation, and overactivity in a CYP-induced cystitis model in rats [214]. Chen et al. revealed that LiESWT significantly alleviated bladder damage and ameliorated inflammation by decreasing the expression of IL-12, TNF-α, NF-κB, and inducible nitric oxide synthase and the level of oxidative stress by reducing NADPH oxidase 1 and NOX-2 expression in a CYP-induced cystitis [215].

Clinically, LiESWT could generate some growth factors or cytokines that inhibit inflammatory reaction and apoptosis, which might have a therapeutic effect on IC/BPS patients. Chuang et al. reported IC/BPS participants treated with LiESWT showed improvement in the VAS pain scores compared to the placebo group in a prospective, randomized, double-blind, placebo-controlled clinical study [216]. Chuang et al. also revealed the LiESWT group exhibited a significant reduction of bladder pain in the OSS and VAS scores compared to the placebo group after 4 weeks treatment, and the effects were persistent at least for 12 weeks. Urinary IL-4, IL-9, and VEGF mediation may be involved in the pathophysiologic mechanisms and response to LiESWT treatment [151]. However, the relationship between the intensity and duration of LiESWT and its therapeutic effects required further evaluation.

### 8.4. Regenerative Medicine Therapy Using Platelet-Rich Plasma (PRP)

PRP is a centrifuged autologous concentrate serum enriched with platelets. Platelets could release cytokines and growth factors stored in alpha granules, thereby stimulating inflammatory cascade, initiating angiogenesis, cell proliferation, and differentiation. These key regenerative growth factors include TGF-β, VEGF, epidermal growth factor (EGF), basic fibroblast growth factor (bFGF), and platelet-derived growth factor (PDGF) lead to tissue healing. PRP has neuroprotective function and can promote nerve regeneration. For example, the effect of local administration of PRP can increase the regenerative axon diameter and number of Schwann cells, which show enhanced nerve regeneration in the rabbit model [217]. The application of PRP regulated tissue regeneration in trauma patients and animal models. PRP injection in burn scar areas for treating burn-induced neuropathic scar pain showed PRP alleviated inflammation and increased the activation of microglia and astrocytes through the activation of p-p38/NF¦ÊB signaling pathways [218,219].

The clinical utilization of PRP has been applied in several urinary diseases, including recurrent bacterial cystitis [220], ED, and IC/BPS [221]. Repeated intravesical injections of autologous PRP have been shown to improve symptoms in patients with IC/BPS characterized by urgency, frequency, and suprapubic pain [221]. Moreover, PRP therapy improved inflammation, increased bladder capacity, and relieved bladder pain in patients with IC/BPS [222]. Donmez et al. showed that PRP instillation significantly stimulated angiogenesis and increased cell proliferation and enhanced bladder mucosal repair in the CYP-induced cystitis rabbit model [220]. PRP modulated inflammatory processes and promoted tissue repair, which have been proved to relieve bladder pain and increase bladder capacity in C/BPS patients [223]. Jhang et al. reported repeated PRP intravesical injections once a month for 4 months, 67.5% of IC/BPS participants improved symptoms and increased functional capacity after 3 month follow-up. In addition, PRP treatment significantly increased urinary IL-2 and IL-8 biomarkers in these IC/BPS patients, suggesting that the effect of PRP through modulating bladder inflammation responses [224,225]. Currently, the therapeutic use of PRP in IC/BPS patients is still experimental.

**Table 4 diagnostics-12-00075-t004:** Assessments of therapeutic outcomes with potential urinary biomarkers for IC/BPS after medical therapies.

Treatment	Doses/ Indication	Therapeutic Efficacy	Urinary Biomarker	Reference
BoNT-A (Neurotoxin)	100–200 IU	Improvement of urinary urgency, bladder pain, and bladder capacity	Reduction of the NGF level	[226]
		Improvement of IC/BPS symptoms	Reduction of the VEGF level	[83]
HA (Glycoprotein)	40 mg	Improvement of bladder pain and storage symptoms and reduction of bladder pain	Reduction of the NGF level	[190,196,197,227]
LiESWT	2000 shocks, frequency of 3 pulses/sec, density of 0.25 mJ/mm^2^ for 4 weeks	Improvement of IC/BPS symptoms Reduced scores of OSS, ICSI, ICPI, and VAS and bladder pain	Reduction of the VEGF level	[151]
	frequency of 3 pulses/sec density of 0.25 mJ/mm^2^ for 4 weeks	Improvement of IC/BPS symptoms Reduced scores of OSS and pain scale	Reduction of the IL-4, IL-9, and VEGF levels	[151,216]
PRP	10 mL extracted from 50 mL of whole blood	Improvement of IC/BPS symptoms and increased functional capacity	Increase of the IL-2 and IL-8 levels	[225]

Note: DMSO, dimethyl sulfoxide; GAG, glycosaminoglycan; HA, hyaluronic acid; ICSI, Interstitial Cystitis Symptom Index; ICPI, Interstitial Cystitis Problem Index; LiESWT, low intensity extracorporeal shock wave therapy; NGF, nerve growth factor; PRP, platelet-rich plasma; VEGF, vascular endothelial growth factor; OSS, O’Leary–Sant symptom scores; VAS, Visual Analog Pain Scale.

## 9. Conclusions

The pathophysiology of IC/BPS is composed of multiple possible factors, such as chronic inflammation, autoimmune disorders, neurogenic hyperactivity, urothelial defects, abnormal angiogenesis, oxidative stress, neurogenic hyperactivity, and exogenous urine substances, which play important roles in the pathophysiology of IC/BPS. Classifications of IC/PBS need to identify new biomarkers for diagnosis, as well as for treatment efficiency in patients with IC/PBS. Therefore, abnormal expressions of several urine and serum specimens, including growth factor, methylhistamine, glycoprotein, chemokine, and cytokines, might be useful as biomarkers for IC/BPS diagnosis. The therapeutic efficacy assessments for IC/BPS are summarized in Figure 1.

## Figures and Tables

**Figure 1 diagnostics-12-00075-f001:**
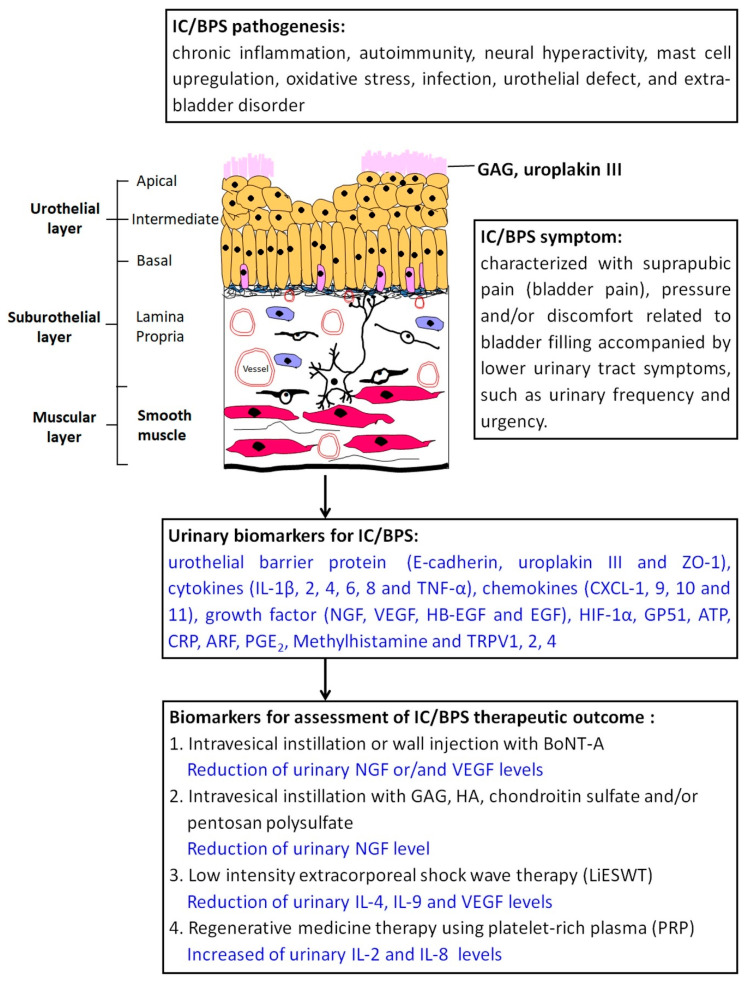
Urinary biomarkers in interstitial cystitis/bladder pain syndrome and its impact on therapeutic outcome. Note: APF, antiproliferative factor; ATP, adenosine triphosphate; CRP, C-reactive protein; CXCR, CXC chemokine receptor; CXCL, CXC chemokine ligand; EGF, epidermal growth factor; FGF, fibroblastic growth factor; GAG, glycosaminoglycan; GP51, glycoprotein 51; HA, hyaluronic acid; HB-EGF, heparin-binding epidermal growth factor; HIF-1α, hypoxia-inducible factor-1-α; IC/BPS, interstitial cystitis/bladder pain syndrome; IL, interleukin; NGF, nerve growth factor; PGE_2_, prostaglandin E_2_; TNF-α, tumor necrosis factor-α; TRPV, TRP vanilloid; VEGF, vascular endothelial growth factor; ZO-1, zonula occludens-1.

**Table 1 diagnostics-12-00075-t001:** Definition, classification, histology, diagnosis, and treatment show differences between HIC/BPS and NHIC/BPS.

Item		HIC/BPS	NHIC/BPS
Definition		IC/BPS with Hunner lesions	IC/BPS without Hunner lesions
Classification		Hunner-type (Ulcerative) type	Non-Hunner-type (Unulcerative) type
Histopathology	Subepithelial chronic inflammation	Present	Absent or minimal
	Types of infiltrating inflammatory cells	Lymphocytes and plasma cells are dominant.	Plasma cells are few.
	Lymphoid follicles	Often present	Extremely rare
	Urothelium	Frequently denuded	Full layer is preserved
	Mast cell	Often present	Extremely rare
Diagnosis	Cystoscopy	Hunner lesions: presence	Hunner lesions: absence
	Bladder capacity	Low	Low
	Bladder biopsy	Dense inflammatory infiltration and epithelial denudation	Slight inflammation
Treatment	Fulguration/Distension	Fulguration/Distension	Distension
	Intravesical instillation	HA, chondroitin sulfate, Botulinum toxin, steroid	HA, chondroitin sulfate, Botulinum toxin, steroid
	Medicine	Necessary	Necessary

**Table 2 diagnostics-12-00075-t002:** Clinical symptom, histopathology, biomarkers, diagnosis, and treatment exhibit differences between IC/PBS and OAB.

Item	IC/PBS	OAB
Clinical symptom	Bladder pain (suprapubic pain), urinary frequency, nocturia, and urgency	Daytime frequency of micturition ≥8 times, nocturia ≥1 times, urgency ≥1 time, or urgency incontinence ≥1 time.
Histopathology	Mast cell infiltration	
Urothelial defects	Present in Hunner-type IC/PBS	Absent or minimal
Biomarkers	The levels of NGF in urine and bladder tissue, serum cytokines, and serum CRP were elevated.	
Diagnosis	Cystoscopy, bladder capacity, 3-day urinary diary	Uroflowmetry, bladder capacity, 3-day urinary diary,
Symptom score	O’Leary–Sant Problem Index (ICSI and ICPI), VAS	OABSS, ICIQ-SF, UDI-6, and IIQ-7
Medical therapy	BoNT-A intravesical injection, LiESWT, PRP	ß3 agonist, BoNT-A intravesical injection, LiESWT

Note: BoNT-A, OnabotulinumtoxinA (botulinum toxin A); CRP, C-reactive protein; IC/BPS, interstitial cystitis/bladder pain syndrome; ICSI, Interstitial Cystitis Symptom Index; ICPI, Interstitial Cystitis Problem Index; ICIQ-SF, International Consultation on Incontinence Questionnaire-Short Form; IIQ-7, Incontinence Impact Questionnaire-7 score LiESWT, Low-intensity extracorporeal shock wave therapy; NGF, nerve growth factor; OAB, overactive bladder; OABSS, Overactive Bladder Symptom Scores; PRP, platelet-rich plasma; UDI-6, Urogenital Distress Inventory-Short Form; VAS, visual analog scale.

## Data Availability

Not applicable.

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
