# Peer review of "Urinary Biomarkers in Interstitial Cystitis/Bladder Pain Syndrome and Its Impact on Therapeutic Outcome"

_diagnostics, 2021, doi:10.3390/diagnostics12010075_

Round 1

Reviewer 1 Report

This review article is a comprehensive summary on all aspects of BPS/ IC. Although such a review has a lot of fruit for thought, it is too comprehensive and it lacks in- depth analysis on every aspect of the disease. It would be wiser to focus on one aspect of the disease such as pathophysiology or available studies on potential biomarkers. Overall the article is well organised, however the English grammar and vocabulary is very poor.

I am particularly concerned about the lack of cohesion between the subsequent sentences in several  paragraphs of the text. For example, page 4 lines starting from 13, here the authors jump from inflammation in BPS/ IC to TLR- 4 receptors in OAB and central sensitisation. This section and several others in the text, read like it was written by a machine rather than someone presenting ideas / facts/ findings in a systematic way.  

More importantly, throughout the text there are several misleading statements that gives the impression that the authors lack the expertise and knowledge that is necessary to synthesize the available knowledge on the field. For example, in the abstract, line 48 authors state that many symptoms of OAB and BPS/ IC overlap and that they could be distinguished using biomarkers, which is totally wrong. Also, in the conclusion section, authors list a few biomarkers that could be used as disease specific diagnostic and prognostic tools! There are no known markers that could accurately be used in clinic as a diagnostic and/ or prognostic tool. Also we are very far from realistically finding ‘disease specific’ biomarkers for BPS/ IC.      

My specific comments are:

  • There are also some ambiguous terminologies used throughout the text such as ‘nerve hyperactivity’, what do the others mean by nerve hyperactivity?
  • Page 6 line 25- what is blood nutrition?
  • Page 6 line 28- what is immunological inflammations? Are there any non- immunological inflammations?
  • Page 15 line 65- This sentence on BONT- A increasing bladder capacity and improving urodynamic findings is totally wrong.
  • Table 4- Here the authors tried to identify a relationship between therapeutic efficacy and urinary biomarkers Although I understand the idea and effort, there is no robust data to suggest a mechanism of action for any drugs used in BPS/ IC and there no known biomarkers for any type of treatment.
  • In section 3.1. Authors need to include the well accepted phenotyping by ESSIC and EAU that uses histologic and cystoscopy findings to classify the subjects into clinical groups.

Author Response

Dear Dr. Jochen Neuhaus,

We wish to thank the Editorial Board for the review of our manuscript entitled " Urinary biomarkers in interstitial cystitis/bladder pain syndrome and its impact on therapeutic outcome", which is being considered by the Diagnostics for publication.

By addressing every comment made by the reviewers, we have revised our manuscript. All the changes made in the manuscript are marked in red font.

We would like to thank you and the Editorial Board for the consideration and the intelligent review of our manuscript, which results in the revised manuscript of better quality.

Sincerely yours,

Yung-Shun Juan MD, PhD.

November 20th, 2021

Reviewer 1

Comments and Suggestions for Authors

This review article is a comprehensive summary on all aspects of BPS/ IC. Although such a review has a lot of fruit for thought, it is too comprehensive and it lacks in- depth analysis on every aspect of the disease. It would be wiser to focus on one aspect of the disease such as pathophysiology or available studies on potential biomarkers. Overall the article is well organized, however the English grammar and vocabulary is very poor.

  1. I am particularly concerned about the lack of cohesion between the subsequent sentences in several paragraphs of the text. For example, page 4 lines starting from 13, here the authors jump from inflammation in IC/BPS to TLR- 4 receptors in OAB and central sensitization. This section and several others in the text, read like it was written by a machine rather than someone presenting ideas / facts/ findings in a systematic way.

Response: We rearranged the related description to make it smoother in the section 3.2.1. Chronic inflammation section (please refer to page 4, line 143-152). All the changes made in the manuscript are marked in red font. We also move the description “Bladder inflammation might induce afferent nerve hyperactivity and result in pain symptoms in IC/BPS. For example, Toll-Like Receptor-4 (TLR-4) has been shown as important factor in central pain sensitization. Interactions between TLR-4 mediated inflammation and sex hormones were considered to be a potential mechanism in the different prevalence of pain condition in female and male IC/BPS patients [27]. Therefore, TLR-4 mediated inflammation was associated with painful symptoms and nerve hyperactivity, such as bladder pain, frequency, and urgency, especially in female IC/BPS patients” to the section 3.2.3. Neurogenic hyperactivity (please refer to page 5, line 179-186).

  1. More importantly, throughout the text there are several misleading statements that gives the impression that the authors lack the expertise and knowledge that is necessary to synthesize the available knowledge on the field. For example, in the abstract, line 48 authors state that many symptoms of OAB and BPS/ IC overlap and that they could be distinguished using biomarkers, which is totally wrong. Also, in the conclusion section, authors list a few biomarkers that could be used as disease specific diagnostic and prognostic tools! There are no known markers that could accurately be used in clinic as a diagnostic and/ or prognostic tool. Also, we are very far from realistically finding ‘disease specific’ biomarkers for BPS/ IC.

Response: Thank you for your suggestion. We have deleted the description “Although symptoms of IC/BPS often overlap with overactive bladder or hypersensitive bladder, several biomarkers may help distinguish them. Besides,” from the Abstract section (please refer to page 1), and modified the related description from ”abnormal expressions of several urine specimens, including …, may serve as disease-specific biomarkers in the diagnostic or prognostic tool for the clinical management in IC/BPS.“ to “Abnormal expressions of several urine  and serum specimens, including …, might be useful as biomarkers for IC/BPS diagnosis”, which has been added to the Abstract section (please refer to page 1, lines 48-50) and Conclusion section (please refer to page 20, lines 783-786).

My specific comments are:

  1. There are also some ambiguous terminologies used throughout the text such as ‘nerve hyperactivity’, what do the others mean by nerve hyperactivity?

Response: Peter A. Smith (2020) indicated that sensory nerve hyperactivity (hyperexcitability) generated by nerve injury, peripheral neuropathy or disease are often expressed as neuropathic pain. This hyperactivity results from modulation, increased activity and /or expression of voltage-gated Na+ channels and hyperpolarization-activated cyclic nucleotide–gated (HCN) channels (Front. Cell. Neurosci., 17 September 2020). Besides, Yoshimura et al., indicated that hyperactivity of C-fiber afferents might lead to pain sensation in response to normal non-noxious distension of the bladder (Yoshimura., 2014). So, we revised nerve hyperactivity (hyperexcitability) to the section 3.2.3. Neurogenic hyperactivity (hyperexcitability) (please refer to page 5, lines 179 and 185) and the section 3.2.4. Urothelial defect section (please refer to page 6, lines 230-231).

The references are listed here:

Peter A Smith. + Channels in Primary Afferents and Their Role in Nerve Injury- Induced Pain. Front. Cell. Neurosci., 17 September 2020,Sep 17;14: 566418. 

Yoshimura, N.; Oguchi, T.; Yokoyama, H.; Funahashi, Y.; Yoshikawa, S.; Sugino, Y.; Kawamorita, N.; Kashyap, M. P.; Chancellor, M. B.; Tyagi, P.; Ogawa, T., Bladder afferent hyperexcitability in bladder pain syndrome/interstitial cystitis. Int J Urol 2014, 21 Suppl 1, 18-25.

  1. Page 6 line 25- what is blood nutrition?

Response: We have modified the related description from” Angiogenesis played a vital role in maintaining blood nutrition and oxygen to supply the regeneration of dysfunction bladder“ to “Angiogenesis played a vital role in maintaining blood vessels delivering nutrition and oxygen to supply the regeneration of dysfunctional bladder”, which has been added to the section 3.2.6. Abnormal angiogenesis section (please refer to page 6, line 275-276).

  1. Page 6 line 28- what is immunological inflammations? Are there any non- immunological inflammations?

Response: We have modified the related description from ”Damage-associated molecular patterns from the degenerated cells by toxic substances might promote immunological inflammations“ to “Damage-associated molecular patterns from the degenerated cells by toxic substances might promote immune response and pathological inflammation”, which has been added to the section 3.2.7. Exogenous urine substances section (please refer to page 7, lines 292-294).

  1. Page 15 line 65- This sentence on BONT- A increasing bladder capacity and improving urodynamic findings is totally wrong.

Response: We have modified the related description “BoNT-A can relieve pain, urgency and frequency for refractory IC/BPS and improve the functional bladder capacity and urodynamic parametersin the section 8.2. Intravesical instillation or injection with BoNT-A section (please refer to page 16, lines 683-684). We also added the descriptions “Kuo et al. suggested that functional bladder capacity (136 ± 79.1 vs. 208 ± 110, p = 0.000) was significant improvement after four repeated injections with BONT- A in IC/BPS patients” (please refer to page 16, lines 684-686).

  1. Table 4- Here the authors tried to identify a relationship between therapeutic efficacy and urinary biomarkers. Although I understand the idea and effort, there is no robust data to suggest a mechanism of action for any drugs used in BPS/ IC and there no known biomarkers for any type of treatment.

Response: Thank you for your suggestion. We have modified the related description from ”Impact of biomarkers“ to “Impact of potential biomarkers”, which has been added to the section 8. Impact of biomarkers for assessment of IC/BPS therapeutic outcome section (please refer to page 15, lines 648 and 649). We also have revised the related description from” Assessments of therapeutic outcomes with urinary biomarkers for IC/BPS after medical therapies“ to “Assessments of therapeutic outcomes with potential urinary biomarkers for IC/BPS after medical therapies”, which has been added to the Table 4 legend (please refer to page 17, line 764).

  1. In section 3.1. Authors need to include the well accepted phenotyping by ESSIC and EAU that uses histologic and cystoscopy findings to classify the subjects into clinical groups.

Response: We have added the descriptions in the section 3.1. Classification section (please refer to page 3, lines 102-110), as ” The Study of Interstitial Cystitis (ESSIC) subtype patients with BPS into grade 1 (normal), grade 2 [with glomerulations grade II (large submucosal bleeding) or grade III (diffuse global mucosal bleeding)], and grade 3 [Hunner lesions (with or without glomerulations)] according to cystoscopy with hydrodistension, and classified into grade A (normal), grade B (with inconclusive), and grade C (histology showing inflammatory infiltrates and/or detrusor mastocytosis and/or granulation tissue and/or intrafascicular fibrosis) according to biopsy diagnosis. The European Association of Urology (EAU) guide-lines further give a recommendation that grade A diagnosis requires hydrodistension and biopsy.“

Reviewer 2 Report

The current diagnosis of IC/BPS is an exclusive diagnosis based on relative symptoms. The development of noninvasive biomarkers  is important  for both for  diagnosis and therapy. This review gives a comprehensive induction about the disease and current investigated biomarkers. It is helpful for readers on further understanding of the disease.

Major problem:

    This review has spent too much time on the introduction of diseases, including etiology, pathogenesis, histopathology, etc.The introduction of biomarkers is relatively weak. There is still no biomarker accepted widely and available  in clinic, the advantages and disadvantages of the markers reported should  be further clarified to give  better guidance to the readers.

Author Response

Dear Dr. Jochen Neuhaus,

We wish to thank the Editorial Board for the review of our manuscript entitled " Urinary biomarkers in interstitial cystitis/bladder pain syndrome and its impact on therapeutic outcome", which is being considered by the Diagnostics for publication.

By addressing every comment made by the reviewers, we have revised our manuscript. All the changes made in the manuscript are marked in red font.

We would like to thank you and the Editorial Board for the consideration and the intelligent review of our manuscript, which results in the revised manuscript of better quality.

Sincerely yours,

Yung-Shun Juan MD, PhD.

November 20th, 2021

Reviewer 2

Comments and Suggestions for Authors

The current diagnosis of IC/BPS is an exclusive diagnosis based on relative symptoms. The development of noninvasive biomarkers is important for both for diagnosis and therapy. This review gives a comprehensive induction about the disease and current investigated biomarkers. It is helpful for readers on further understanding of the disease.

Major problem:

This review has spent too much time on the introduction of diseases, including etiology, pathogenesis, histopathology, etc. The introduction of biomarkers is relatively weak. There is still no biomarker accepted widely and available in clinic, the advantages and disadvantages of the markers reported should be further clarified to give better guidance to the readers.

Response: Thank you for your suggestion. We have added the description “Taken together, the potential biomarkers of urothelial barrier protein (Uroplakin III, E-Cadherin and ZO-1), Apoptotic signaling molecules (Bad, Bax, and Cleaved caspase-3), HIF-1α, and TRPV1, 2, 4 need to identify from bladder biopsy and further analysis by real-time PCR for RNA expression and by western blot or immunohistochemistry stain for protein expression. The other potential biomarkers of proinflammatory cytokines, chemokines and proteins (CXCL-1, CXCL-9, CXCL-10, CXCL-11, IL-1β, IL-2, IL-4, IL-6, IL-8, TNF-α and IgE), growth factors (NGF, VEGF, HB-EGF, EGF and APF), GP51, ATP, CRP, methylhistamine, PGE2, and platelet derived endothelial cell growth factor/thymidine phosphorylase (PDECGF/TP) can be analysis from urine supernatant samples and serum samples and further analysis by Enzyme-linked immunosorbent assay for suspended protein expression, which is more rapid, popular, easy and noninvasive than bladder biopsy analysis. In addition, the urine proteome showed a better association with IC/PBS symptoms than the serum proteome.” in the section 7. Current investigated biomarkers in IC/BPS section (please refer to page 14, lines 628-640).

Reviewer 3 Report

The manuscript is an interesting review of the pathophysiology, histopathology, clinical diagnosis, and current interstitial cystitis/bladder pain syndrome biomarkers. It is well written and organized. 

I consider it a relevant topic as the causes of this entity are not well explained, and the diagnosis and treatment are difficult. Having more information about this disease could improve the management of IC/BPS.

The conclusions are consistent and address the central question of the review, which is to provide some clarity on the management and diagnosis of IC/BPS.

My only suggestion is the following - please include data from the following articles:

Jiang, YH., Jhang, JF., Hsu, YH. et al. Urine biomarkers in ESSIC type 2 interstitial cystitis/bladder pain syndrome and overactive bladder with developing a novel diagnostic algorithm. Sci Rep 11, 914 (2021). https://doi.org/10.1038/s41598-020-80131-5

Charrua, A., Mendes, P. & Cruz, C. Biomarkers for Bladder Pain Syndrome/Interstitial Cystitis. Curr Bladder Dysfunct Rep 16, 12–18 (2021). https://doi.org/10.1007/s11884-020-00626-9

Author Response

Dear Dr. Jochen Neuhaus,

We wish to thank the Editorial Board for the review of our manuscript entitled " Urinary biomarkers in interstitial cystitis/bladder pain syndrome and its impact on therapeutic outcome", which is being considered by the Diagnostics for publication.

By addressing every comment made by the reviewers, we have revised our manuscript. All the changes made in the manuscript are marked in red font.

We would like to thank you and the Editorial Board for the consideration and the intelligent review of our manuscript, which results in the revised manuscript of better quality.

Sincerely yours,

Yung-Shun Juan MD, PhD.

December 18th, 2021

Dr. Reviewer 3

Comments and Suggestions for Authors

The manuscript is an interesting review of the pathophysiology, histopathology, clinical diagnosis, and current interstitial cystitis/bladder pain syndrome biomarkers. It is well written and organized.

I consider it a relevant topic as the causes of this entity are not well explained, and the diagnosis and treatment are difficult. Having more information about this disease could improve the management of IC/BPS.

The conclusions are consistent and address the central question of the review, which is to provide some clarity on the management and diagnosis of IC/BPS.

My only suggestion is the following - please include data from the following articles:

Jiang, YH., Jhang, JF., Hsu, YH. et al. Urine biomarkers in ESSIC type 2 interstitial cystitis/bladder pain syndrome and overactive bladder with developing a novel diagnostic algorithm. Sci Rep 11, 914 (2021). https://doi.org/10.1038/s41598-020-80131-5

Charrua, A., Mendes, P. & Cruz, C. Biomarkers for Bladder Pain Syndrome/Interstitial Cystitis. Curr Bladder Dysfunct Rep 16, 12–18 (2021). https://doi.org/10.1007/s11884-020-00626-9

Response: Thank you for your suggestion. We have added the description and reference in the section 7. Current investigated biomarkers in IC/BPS section as “In recent years, the black box representing BPS/IC has expanded from the bladder-centred concept to the whole body (Charrua, Mendes et al. 2021)……  To identify the key molecules by using adequate sample size and selecting controls for in IC/BPS will help to improve the efficacy of treatment and identify biomarkers of the dis-ease (Charrua, Mendes et al. 2021).” (please refer to page 10, lines 427-435). We also added the description and reference in the section 7. Current investigated biomarkers in IC/BPS section as “Besides, urine CXCL-10 is elevated in patients with IC/BPS, but not in OAB patients (Jiang, Jhang et al. 2021). The up-regulated levels of serum TNF-α, IL-1β, 6, and 8, and urine CXCL-10 level in IC/BPS patients might help provide as an appropriate diagnostic tool.” (please refer to page 11, lines 475-478) and Table 3.  

We also added references in manuscript:

  1. Charrua, A., P. Mendes, and C. Cruz, Biomarkers for Bladder Pain Syndrome/Interstitial Cystitis. Current Bladder Dysfunction Reports, 2021. 16(1): p. 12-18.
  2. Jiang, Y.H., et al., Urine biomarkers in ESSIC type 2 interstitial cystitis/bladder pain syndrome and overactive bladder with developing a novel diagnostic algorithm. Sci Rep, 2021. 11(1): p. 914.

Round 2

Reviewer 2 Report

The revised manuscript added an conclusion on the significance of biomarkers on the diagnosis and evaluation of therapeutic outcome. The manuscript can be accepted in present form. 

Author Response

Dear Dr. Jochen Neuhaus,

We wish to thank the Editorial Board for the review of our manuscript entitled " Urinary biomarkers in interstitial cystitis/bladder pain syndrome and its impact on therapeutic outcome", which is being considered by the Diagnostics for publication.

By addressing every comment made by the reviewers, we have revised our manuscript. All the changes made in the manuscript are marked in red font.

We would like to thank you and the Editorial Board for the consideration and the intelligent review of our manuscript, which results in the revised manuscript of better quality.

Sincerely yours,

Yung-Shun Juan MD, PhD.

December 18th, 2021

Reviewer 2

Comments and Suggestions for Authors

The revised manuscript added an conclusion on the significance of biomarkers on the diagnosis and evaluation of therapeutic outcome. The manuscript can be accepted in present form.

Response: Thank you for your comments.
